# Diabetic Glycation of Human Serum Albumin Affects Its Immunogenicity

**DOI:** 10.3390/biom14121492

**Published:** 2024-11-23

**Authors:** Cresci-Anne C. C. Croes, Marialena Chrysanthou, Tamara Hoppenbrouwers, Harry Wichers, Jaap Keijer, Huub F. J. Savelkoul, Malgorzata Teodorowicz

**Affiliations:** 1Department of Cell Biology and Immunology, Wageningen University and Research Centre, 6700 AH Wageningen, The Netherlandsgosia.teodorowicz@wur.nl (M.T.); 2Department of Food Quality and Design, Wageningen University and Research Centre, 6708 WG Wageningen, The Netherlands; marialena.chrysanthou@wur.nl (M.C.); tamara.hoppenbrouwers@wur.nl (T.H.); 3Department of Food Chemistry, Wageningen University and Research Centre, 6700 AA Wageningen, The Netherlands; harry.wichers@wur.nl; 4Department of Food and Biobased Research, Wageningen University and Research Centre, 6700 AA Wageningen, The Netherlands; 5Department of Human and Animal Physiology, Wageningen University and Research Centre, 6700 AH Wageningen, The Netherlands; jaap.keijer@wur.nl

**Keywords:** diabetes, AGEs, HSA, RAGE, macrophages, inflammation, receptors, methylglyoxal

## Abstract

Advanced glycation end-products (AGEs) are products of a non-enzymatic reaction between amino acids and reducing sugars. Glycated human serum albumin (HSA) increases in diabetics as a consequence of elevated blood glucose levels and glycating metabolites like methylglyoxal (MGO). The impact of different types of glycation on the immunomodulatory properties of HSA is poorly understood and is studied here. HSA was glycated with D-glucose, MGO, or glyoxylic acid (CML). Glycation-related biochemical changes were characterized using various biochemical methods. The binding of differentially glycated HSA to AGE receptors was determined with inhibition ELISAs, and the impact on inflammatory markers in macrophage cell line THP-1 and adherent monocytes isolated from human peripheral blood mononuclear cells (PBMCs) was studied. All glycation methods led to unique AGE profiles and had a distinct impact on protein structure. Glycation resulted in increased binding of HSA to the AGE receptors, with MGO modification showing the highest binding, followed by glucose and, lastly, CML. Additionally, modification of HSA with MGO led to the increased expression of pro-inflammatory markers in THP-1 macrophages and enhanced phosphorylation of NF-κB p65. The same pattern, although less prominent, was observed for HSA glycated with glucose and CML, respectively. An increase in pro-inflammatory markers was also observed in PBMC-derived monocytes exposed to all glycated forms of HSA, although HSA–CML led to a significantly higher inflammatory response. In conclusion, the type of HSA glycation impacts immune functional readouts with potential relevance for diabetes.

## 1. Introduction

The increase in patients with diabetes mellitus type 2 (T2DM) is a concern in modern society [1]. Patients exhibit chronic hyperglycemia meaning constantly elevated levels of glucose in the circulation. This circulating glucose can react with free amino groups of serum proteins, including human serum albumin (HSA), through a dose-dependent, non-enzymatic reaction called glycation (Maillard reaction) [2,3,4]. HSA is particularly prone to glycation due to its long half-life and high blood concentration [5,6]. Its glycation mainly occurs on free lysine and arginine residues [5,6] and consists of three steps: the formation of a Schiff base, followed by the formation of Amadori products, and subsequent degradation to reactive dicarbonyls. These dicarbonyls then react with amino acids to form advanced glycation end-products (AGEs) [7]. Glycated HSA ranges from 1 to 10% of total plasma HSA in healthy individuals and to 20–50% in diabetic patients [5,8]. This increased blood glucose level and glycation in T2DM leads to AGEs such as N ε-(carboxymethyl)lysine (CML) and N ε-(carboxyethyl)lysine (CEL) [9,10]. However, circulating levels of other glycating metabolites like the dicarbonyl methylglyoxal (MGO), a byproduct of glycolysis, also increase in individuals suffering from T2DM and can lead to the formation of AGEs like MGO-derived hydroimidazolone (MG-H1) [11]. In conditions like increased metabolic stress, glycation involving MGO has been documented in individuals suffering from T2DM [11].

Although glucose and MGO have been reported to increase in parallel following a glucose tolerance test [12], the biological effects may be different since MGO glycation occurs at a much faster rate compared to glucose glycation [13,14]. In healthy individuals, the body is able to detoxify MGO via the glyoxalase system; however, this detoxification system is also impaired in T2DM [15]. Glycation of proteins can have negative implications on the body in two main ways. First, glycation alters the structure of proteins through processes like protein crosslinking and aggregation, therefore modifying their function [16], i.e., transporter function or antioxidant properties of HSA [17] may be affected by glycation. Second, aggregated and glycated proteins can interact with receptors expressed on immune cells, leading to an inflammatory response and oxidative stress [18,19]. AGEs were shown to interact with receptors such as receptors for advanced glycation end-products (RAGE), Galectin-3, CD36, and SR-A1 [20].

RAGE is the best-known and most studied receptor for AGEs [7,21] and was shown to be the primary receptor leading to AGE-induced inflammatory responses and oxidative stress [22,23]. RAGE, as a pattern-recognition receptor, belongs to the immunoglobulin superfamily of cell surface molecules [16,24] and is expressed by many cell types, including immune cells such as monocytes and macrophages. RAGE can also interact with aggregated proteins [20,25,26], and its activation is implicated in the onset of several diseases, including T2DM [27]. There are numerous AGE structures present in plasma, but they exist at very low concentrations, making it unlikely that they significantly influence RAGE signaling [28]. However, increased blood plasma concentrations of MGO and CML have both been associated with the upregulation of RAGE expression [29,30]. Research indicates that glucose and CML modifications primarily occur at lysine residues, whereas modifications with MGO occur at both lysine and arginine residues [3]. Upon modification of either lysine or arginine residues on HSA, the positive charge on these residues is removed, increasing the affinity of glycated HSA for RAGE [31].

Few studies suggest that glycated HSA can induce pro-inflammatory responses in immune cells [18,19,32,33]. A direct comparison between glucose-glycated HSA and MGO-glycated HSA is, however, limited. The limited number of studies investigating the functional and immunogenic impact of different modifications are mostly outdated [34,35], using bovine serum albumin instead of HSA [36,37,38,39,40], glycation method poorly described [33,41,42,43], ignore endotoxin levels [41], or have incomparable readouts [33,39,44,45,46,47,48], resulting in inconsistent findings. The structure-function relationship of HSA glycated by different metabolites remains poorly understood, although BSA studies suggest that both the structural properties of the protein (i.e., beta-sheet number) and chemical structures of AGE-modification are crucial for binding to AGE receptors [3]. Understanding the impact of endogenous AGEs on inflammation is relevant, given the link between protein glycation and diabetic pathology.

Thus, the aim of this study was to investigate how different glycation methods of HSA, specifically glucose vs. MGO, correlate with immunomodulatory effects. We hypothesized that AGEs generated from glycation of HSA with glucose and MGO uniquely alter HSA structure, impacting binding to known AGE receptors and macrophage activation. Additionally, HSA with specific CML modification was taken along as an important marker for glycation with glucose. A deeper understanding of the impact of different glycations could enhance our understanding of the onset or progression of diseases such as T2DM.

## 2. Materials and Methods

### 2.1. Endotoxin Purification

Protein samples were purified from endotoxin prior to glycation and cell experiments using the Triton X-114 method, as described by Teodorowicz et al. [49]. In short, TX-114 was added to the HSA and soy protein solution (10 mg/mL) to a final concentration of 2% *v*/*v*. The samples were then incubated at 4 °C for 30 min on a tumbler. The samples were then incubated at 37 °C for 10 min, followed by centrifugation at 20,000× *g* or 20 min at 37 °C. The top part containing the protein was separated from the Triton X-114 layer by pipetting. Next, Bio-Beads SM-2 (Biorad cat. #1528920, Hercules, CA, USA) with high affinity to Triton were added to the collected supernatant and incubated overnight at 4 °C on a tumbler at a ratio of 1 g of beads per 0.07 g Triton X-114, with an assumption of 90% Triton X-114 removal after centrifugation step. Endotoxin level was then determined using the ENDONEXT measuring kit (bioMérieux, Marcy-l’Étoile, France) according to the manufacturer’s instructions.

### 2.2. Proteins and Glycations

Human serum albumin (HSA) (A3782) was purchased from Sigma Aldrich (St. Louis, MO, USA). HSA was glycated under wet conditions in three different ways with high concentrations of glycating agents to ensure maximum glycation. For the glucose glycation, HSA (40 mg/mL) was incubated with 250 mM D-glucose, 200 U penicillin, 200 μg/mL streptomycin, 80 μg/mL gentamycin, and 1.5 mmol/L PMSF at 37 °C for 60 days. For the glycation with methylglyoxal (MGO), HSA (30 mg/mL) was incubated with 100 mM MGO solution (M0252, Sigma) at 37 °C for 24 h. For the CML modification, 15 mg of HSA was incubated with 3.3 mg of glyoxylic acid (G10601, Sigma) and 8.25 mg of sodium cyanoborohydride at 40 °C for 20 h. To filter out residual glycating agent and sodium cyanoborohydride, all samples were filtered using an Amicon 10 kDa filter. For all three types of glycation, a control was included that did not include the glycating agent. All HSA glycations were performed in sterile 4 mL glass vials.

Glycated soy protein was included in a number of experiments. Soy protein extract was retrieved from soy flour using the protocol described by L’Hocine et al. [50]. In short, 20 g of soy flour (Sigma Aldrich S9633) was suspended in heated water (55 °C) at a ratio of 1:10 with pH adjustment and stirring. After centrifugation (30 min, 9000× *g*, 4 °C), the resulting supernatants were pooled, pH adjusted, and subjected to additional centrifugation (20 min, 9000× *g*, 10 °C). The precipitate was washed, suspended in PBS, and centrifuged again, followed by the addition of sodium azide to a concentration of 0.02%. Soy protein extract was then glycated at a 1:1 volume ratio with glucose at 100 °C for 90 min.

### 2.3. O-phtaldialdehyde (OPA) Assay

The O-phtaldialdehyde (OPA) reaction protocol to detect free amino groups on HSA was performed as described by Nielsen et al. [51]. In short, a standard curve was prepared using L-leucine. The OPA reagent consisted of sodium tetraborate (0.10 M), SDS (3.5 mM), OPA (6.0 mM), and DTT (5.7 mM). The analysis was performed in quadruplicate in a 96-well plate. L-leucine and samples were added to the wells, followed by the addition of freshly prepared OPA reagent followed by 20 min incubation at RT. The absorbance was measured at 340 nm in a Spectramax iD3 (Molecular Devices, San Jose, CA, USA), and the amount of free amino groups in the samples was calculated using the standard curve obtained with L-leucine.

### 2.4. Gel Electrophoresis

For sodium dodecyl sulfate polyacrylamide gel electrophoresis (SDS-PAGE), samples (8 μg protein) were prepared by adding 6× sample buffer containing dithiothreitol (DTT), followed by heating at 95 °C for 5 min. TGX 4–15% polyacrylamide gels (Biorad, Hercules, CA, USA) were used and placed in the electrophoresis cell, filled with 1× running buffer (3 g Tris-base, 14 g glycine, 1 g SDS) at 120 V for approx. 45 min. For gel staining, the gel was transferred to a tray with demi water and washed multiple times. GelCode Blue Stain Reagent (Thermo Fisher Scientific, Waltham, MA, USA) was added and incubated for 1 h or overnight. Subsequently, the gel was washed with demi water until destained. Gel was imaged using a Biorad ChemiDoc Imaging System (Biorad, Hercules, CA, USA). Precision Plus Protein Dual Color Standards (Biorad, Hercules, CA, USA) were used for the molecular weight ladder.

For the NATIVE-PAGE electrophoresis, samples were prepared by adding 1 part sample and 2-part native sample buffer consisting of 62.5 mM Tris hydrochloride (pH 6.8), 25% glycerol and 0.01% bromophenol blue and was not heated. TGX 4–15% polyacrylamide gels were used and placed in the electrophoresis cell, filled with 1× native running buffer consisting of 25 mM Tris and 192 mM glycine at 120 V for approx. 45 min. Gel staining was performed identically as with the SDS-PAGE.

### 2.5. Cell Models

THP-1 cells were obtained from ATCC (American Type Culture Collection, Manassas, VA, USA). Cells were cultured in RPMI 1640 medium (supplier) containing 250 mM Hepes, L-Glutamine, 10% FBS, and 1% Pen/strep. Cells were passaged twice per week and were discarded at passage 25+. THP-1 cells were differentiated into macrophages in 24-well plates in a concentration of 1 × 10^6^/mL by stimulation with 10 ng/mL of phorbol 12-myristate-13-acetate (PMA) for 48 h, followed by washing and resting for another 48 h. Cells were then utilized for experiments (see next section).

PBMC was extracted from buffy coats (50 mL) from healthy blood donors (Sanquin Blood Bank, Nijmegen, The Netherlands) using the Ficoll Method. In short, a Leucosep tube is filled with 15 mL of Ficoll Plague Plus separation medium and centrifuged at 1000× *g* for 30 s at RT to position the separation medium below the porous barrier. The Buffy coat is diluted at a 1:1 ratio with sterile warm PBS and mixed by inversion. The diluted blood is then poured into the corresponding Leucosep tubes, which are centrifuged for 10 min at 1000× *g* at RT with the brakes switched off. After centrifugation, the enriched cell fraction (PBMCs) and diluted plasma are transferred to a newly labeled 50 mL tube. The PBMCs are washed twice by adding PBS, centrifuging (7 min, 250× *g*, brakes on), removing the supernatant, resuspending the PBMCs in PBS, and repeating the centrifugation step. After the second wash, the PBMCs are resuspended in 1 mL of RPMI-1640 medium and counted before further use for monocyte isolation. Monocytes were isolated either using the EasySep Human Monocyte Isolation kit (Stemcell technologies #19359, Vancouver, BC, Canada) (RNA experiments) or by adhesion to a plate (Western blot experiments). Cells were seeded at 1 × 10^6^ cells per 1 ml in a 12-well plate with RPMI 1640-Glutamax medium containing 10% FBS, 1% MEM non-essential amino acids, 1% Na-Pyruvate, 1% Pen/strep. The adherent monocytes were utilized for experiments (see next section).

For RNA isolation and qPCR experiments, differentiated THP-1 macrophages or adherent primary monocytes were stimulated with glycated HSA at 100 μg/mL, unless indicated differently, at different time points (3 h, 8 h, 24 h). For Western blot experiments, differentiated THP-1 or adherent primary monocytes were treated with glycated HSA at 100 μg/mL for 30 min, followed by whole-cell lysing.

### 2.6. Blotting

For the dot blot, the nitrocellulose membrane was prepared and cut to the appropriate size. A total of 5 μL of the sample was dotted onto the membrane to a final concentration of 30 μg, followed by a blocking step with 3% BSA in TBS-T (Tween-20 0.1%). SDS-PAGE gels were electroblotted on 0.45 μM NC nitrocellulose blotting membrane (Cytiva, Amersham, UK) with semi-dry Western blot buffer at 15 V for 35 min in a Biorad Transblot SD semi-dry transfer cell (Biorad, Hercules, CA, USA). Except for RAGE Western blotting, membranes were washed 2 × 5 min with 1 × TBS-T and incubated for 1 h at RT in 5% milk powder in TBS-T. The membrane was then washed for 2 × 5 min with TBS-T and then blocked with 3% BSA in TBS-T. After blocking, both the dot blot and Western blot membrane were washed 3 × 5 min, followed by incubation with 1 in 1000 of either anti-CML or anti-MG-H1 antibody in 3% BSA in TBS-T at 4 °C overnight. The following day, both membranes were incubated with 1 in 1000 detection antibody, Goat Anti-mouse Immunoglobulins/HRP (affinity isolated) (Dako, Glostrup, Denmark) and Goat Anti-Rabbit IgG Antibody, HRP-conjugate (Merck, Rahway, NJ, USA), in 3% BSA in TBS-T for 1 h at RT. Subsequently, the membrane was washed for 3 × 5 min with TBS-T. Then WesternBright ECL detection reagent (Advansta, San Jose, CA, USA) was added for 2 min and visualized with chemiluminescence using a Biorad Chemidoc and Image Lab Software Version 6.1 (Biorad, Hercules, CA, USA). For RAGE Western blotting, membranes were washed 2 × 5 min with 1× TBS-T and incubated for 1 h at RT in 5% milk powder in TBS-T. Subsequently, membranes were washed 3 × 5 min with TBS-T followed by incubation with 0.25 μg/mL recombinant human RAGE Fc Chimera protein (Bio-Techne, Minneapolis, MN, USA) at 5% milk powder in TBS-T overnight in 4 °C. The following day, the membrane was washed 3 × 5 min, followed by 1 h incubation in detection antibody Goat Anti-Human IgG-HRP at a concentration of 1:2000 (Southern Biotech, Birmingham, AL, USA) in 5% milk powder in TBS-T. Subsequently, the membrane was washed for 3 × 5 min with TBS-T. Then, the membranes were visualized, as mentioned previously.

Cells were lysed using commercial RIPA buffer supplemented with 1 × protease/phosphatase inhibitor (Cell Signaling, Danvers, MA, USA), followed by measurement of concentration on an Implen NanoPhotometer (Implen GmbH, Munich, Germany). Whole-cell lysate was then prepared for SDS gel electrophoresis as described previously. Subsequently, the gel, membrane, and blot paper were equilibrated in Western blot Semi-Dry Blot Buffer for 30 min, stacked inside of a Transblot SD Semi-Dry Transfer Cell, and ran at 15 V for 35 min. After, the membrane was washed twice with TBS-T, followed by blocking with 5% milk in TBS-T for 1 h. The membrane was then washed 3×, followed by incubation with primary antibody (Phospho-NF-κB p65 (Ser536) (93H1) Rabbit mAb, Cell Signaling) overnight at 4 °C. Next, the membrane was washed 3x in TBS-T, followed by incubation with detection antibody for 1 h at RT. Finally, the membrane is washed 3× in TBS-T, followed by detection by means of ECL with a BioRad Chemidoc. Protein expression was quantified using the ImageLab software. Protein expression is normalized to β-Actin Antibody (Cell Signaling, #4967). Original figures can be found in Appendix A. 

### 2.7. Liquid Chromatography-Mass Spectrometry (LC-MS)

uHPLC-ESI-MS/MS (LC-MS) was used to quantify Furosine, Nε-carboxymethyl lysine (CML), and carboxyethyl lysine (CEL). To ensure accuracy, Solid Phase Extraction (SPE) recovery checks were conducted on samples, including three quality controls. Each sample, set to a concentration of 6 mg/mL, underwent heating with 1 mL 6 M hydrochloride at 110 °C in glass heating tubes for 22 h. Next, 400 μL of the resulting solution was transferred to an MS vial and dried in a vacuum concentrator (SpeedVac, Thermo Fisher Scientific, Waltham, MA, USA) and then reconstituted in a mixture of 500 μL acetonitrile and MilliQ water (1:1 *v*/*v*). Subsequently, the solution was spiked with furosine-d4, CML-d4, and CEL-d4, bringing the final concentration to 500 ng/mL. The analysis was conducted using a Kinetex 2.6 u HILCI 100A, 100 × 2.1 mm column (Phenomenex, Torrance, CA, USA) at a temperature of 35 °C. The elution process involved ultrapure water with 0.1% formic acid (eluent A), acetonitrile with 0.1% formic acid (eluent B), and 50 mM ammonium formate (eluent C). The flow rate was maintained at 0.4 mL/min following the gradient: (0/80/10), (0.8/80/10), (3.5/40/10), (6.5/80/10), (8.0/80/10), (11/80/10). Ionization was set to a positive mode, with a spray voltage of 3500 °C, vaporizing temperature of 250 °C, and sheath gas pressure of 60 psig. The capillary temperature was kept at 290 °C.

### 2.8. Receptor Binding

The binding affinity of the receptors RAGE, Galectin-3, SR-A1, and CD36 to the glycated HSA samples was determined using inhibition ELISA following the protocol described by Zenker et al. [26], with some modifications. In short, the ELISA Maxisorp plate was coated using the glycated soy protein at 20 μg/mL and incubated overnight at 4 °C. Protein concentration was adjusted to a range between 5 and 80 μg/mL depending on the receptor tested, and the optimal concentration of sample selected out of a dilution curve from preliminary data. Prior to the addition to the ELISA plate, the samples were pre-incubated with 0.25 μg/mL of recombinant human RAGE Fc Chimera protein, 1 μg/mL of recombinant human Galectin-3 CF, 0.5 μg/mL of recombinant human SR-A1 protein CF, and 0.25 μg/mL of recombinant Human CD36 chimera protein CF (Bio-Techne, Minneapolis, MN, SUA) on a 1:1 ratio *v*/*v* for 45 min at 37 °C on a Nunc polystyrene plate from Thermo Fisher Scientific (Waltham, MA, USA). In parallel, the ELISA plate was blocked with 3% BSA in PBS for 1 h at RT, followed by washing with PBS-T (0.05% tween). The washing step was repeated after each step of the ELISA procedure. Subsequently, the pre-incubated receptor/sample mixture was transferred to the ELISA plate and further incubated for 1 h at 37 °C. After washing, Goat Anti-Human IgG-HRP (Southern Biotech, Birmingham, AL, USA) (for RAGE and CD36), MSR1 antibody (HRP) (Biorbyt, Cambridge, UK) (for SR-A1) or Human Galectin-3 antibody (Bio-Techne, Minneapolis MN, USA) was added to the plate and incubated on a shaker for 30 min at RT. After washing, the signal was detected using Neogen Enhanced K Blue TMB (Neogen Diagnostics, Lexington, KY, USA) and measured at 450 nm with a reference wavelength of 620–650 nm using a Spectramax iD3 (Molecular Devices, San Jose, CA, USA). Each sample was measured in duplicate, and the experiment was individually repeated 3 times. A positive control of glycated soy and a negative control of ovalbumin or bovine serum albumin were included. The inhibition was calculated using the following formula:Inhibition (%) = (Abs_max_ − (Abs_sample_ − Abs_min_))/Abs_Max_ × 100(1)
where Abs_Max_ represents the absorbance obtained from the receptor without any competition agent, Abs_Min_ represents the absorbance obtained from the blank sample without the receptor, and Abs_sample_ represents the absorbance obtained from the mixture of receptor and each sample. A higher inhibition percentage indicates a stronger binding affinity to the receptor.

### 2.9. Gene Expression

For RNA isolation, cells were washed with PBS, followed by the addition of RLT buffer containing 1% B-mercaptoethanol directly in the well, followed by passing the sample through a needle fitted to a syringe. Then, 350 μL of 70% ethanol is added to the homogenized lysate, and the total volume of 700 μL added to the RNeasy mini column, followed by centrifugation for 30 s at 10,000 rpm. Subsequently, 350 μL RW1 buffer is added to the column, followed by centrifugation. Then 80 μL of DNase solution is added to the column and incubated for 15 min at RT. Another 350 μL of RW1 is then added and centrifuged. The column is then washed 2× with RPE buffer and then eluted with 30 μL of RNase-free water. RNA concentration is measured on the nanodrop. cDNA is synthesized using the Superscript III Reverse Transcriptase kit (Invitrogen, Waltham, MA, USA) according to the manufacturer’s instructions with a Biometra TProfessional Thermocycler (Biometra, Gottingen, Germany), using a total of 250–500 ng RNA per sample. For qPCR analysis, 1× Absolute qPCR SYBR Green Mix (thermoscientific, Waltham, MA, USA), 3 μM of forward and reverse primers, and 1 μg of cDNA were mixed per tube to be analyzed. Samples are run in a Qiagen Rotor-Gene Q (Qiagen, Hilden, Germany). The primers used in this paper are listed in Appendix A. Gene expression was calculated using the data retrieved from the Rotor-Gene Q software Version 2.3.5. and transformed to fold change using the Pfaffle method [52]. Pumilio RNA binding family member 1 (*PUM1*) was determined to be the most stable housekeeping gene after optimization, and therefore data are normalized to the PUM1 gene.

### 2.10. Receptor Detection on Flow Cytometer

To prepare cells for receptor verification via the flow cytometer, THP-1 cells and primary monocytes were seeded in a 96-well plate containing 10^5^ cells in FACS buffer. The plate was centrifuged at 400× *g* for 2 min and incubated with 50 μL of primary antibodies (2.5 ug/mL anti-RAGE (Abcam, ab37647, Cambridge, UK) and 1 µg/mL anti-Galectin-3) for 20 min on ice. Subsequently, the cells were centrifuged at 400× *g* for 2 min, followed by incubation with 50 μL secondary antibodies (Alexa Fluor 488 goat anti-rabbit IgG (Invitrogen, Waltham, MA, USA) and Alexa Fluor 488 goat anti-mouse IgG (Invitrogen)) and incubated for 20 min on ice, followed by resuspension in 150 μL FACS buffer. Hereafter, the fluorophore level was measured by flow cytometry. Data were analyzed using the Flow Jo software Version 10.10.0.

### 2.11. Statistics

For statistical analysis, GraphPad Prism 9.5.1. was used. For multiple sample comparison one-way analysis of variance (ANOVA) was used. For comparison between the two samples, a Student’s *t*-test was used. Results were considered statistically significant at *p* < 0.05.

## 3. Results

### 3.1. Analysis of the Glycation Profile of the Modified HSA Samples

In this study, HSA was glycated using three different glycation methods: D-glucose (HSA–glucose), methylglyoxal (HSA–MGO), and glyoxylic acid (HSA–CML). Prior to glycation, HSA was depleted of LPS contamination and tested for cytotoxicity (Appendix A).

To confirm that HSA was glycated by the methods chosen in this study, we first measured the loss of amino groups via OPA assay, followed by the detection of AGEs on dot blot and LC-MS. The OPA assay shows a significantly decreased fold change in free amino groups in the glycated HSA compared to the non-glycated HSA for all three types of glycation conducted (Figure 1A). The greatest loss in amino groups is seen in HSA–CML, followed by HSA–glucose, and lastly, HSA–MGO. Two dot blots were performed, one against CML (Figure 1B) and the other against MG-H1 (Figure 1C). The anti-CML dot was quantified and showed the highest signal for HSA–CML, followed by HSA–glucose, and no signal for HSA–MGO (Figure 1D). The anti-MG-H1 dot blot showed a high signal for HSA–MGO (Figure 1E). The results of the LC-MS confirm that the three methods of glycation led to the generation of intermediate Maillard reaction products (MRPs) and AGEs. Furosine, a byproduct of early glucose glycation, was the highest in HSA glycated with glucose. CML was found in both HSA–CML and HSA–glucose, with HSA–CML containing more than 10× more CML compared to HSA–glucose. CEL was found to be present in HSA–MGO and HSA–glucose, with the highest in HSA–MGO, roughly 10× higher than HSA–glucose (Table 1). The LC-MS data are also in line with the results from the anti-CML dot blot. These results confirm that the three glycations led to the successful modification of HSA and that each method leads to a different glycation profile.

### 3.2. Glycation Methods Uniquely Impact the Physical Chemical Properties of HSA in SDS-PAGE and NATIVE-PAGE Gels

To gain additional insight into the effect of glycation on HSA, we analyzed the glycated HSAs and their non-glycated HSA controls on SDS- and NATIVE-PAGE. These showed a clear change in protein separation profiles for all the glycated HSAs compared to the non-glycated HSAs (Figure 2A,B). Glucose glycation led to a more smeared appearance with bands at the higher molecular weight (~100–200 kDa), indicating a higher number of aggregates in the sample. Additionally, the main HSA band (~67 kDa) also migrated slower, indicating a higher molecular weight. MGO modification significantly altered the appearance of the protein, as the higher molecular weight bands showed a heavy smear (~100–200 kDa). In HSA–MGO we observed the highest degree of aggregation, prominently visible at the top of the gel. Additionally, the main HSA band (~67 kDa) migrated slower. The CML modification showed a similar profile to the control sample in the SDS gel, except that the main HSA band (~67 kDa) migrated less far, indicating a slightly increased molecular weight. The NATIVE-PAGE results showed clear differences in protein separation in glycated versus non-glycated HSAs, and here, the CML modification showed a clear difference in migration profile (Figure 2B). On the NATIVE-PAGE, all glycated samples migrated further on the gel, with CML migrating furthest, followed by MGO, and lastly, glucose, indicating a lower net positive charge.

To identify the bands containing CML and MG-H1, Western blots were performed after denaturing and native gel electrophoresis (Figure 2C,E). Under denaturing conditions, CML is detected in both glucose-glycated and CML-modified HSA. While the CML signal corresponds to bands seen on HSA–glucose on Coomassie staining, CML-modified HSA shows additional structures at the higher molecular weight (~150–250 kD) on the Western blot, which is not visible with the Coomassie staining. Additionally, CML detection is significantly higher in HSA–CML compared to HSA–glucose (Figure 2D). The MG-H1 signal seen under denaturing conditions corresponds to the bands seen on the Coomassie for HSA modified with MGO (Figure 2C,D). Under native conditions, HSA–CML is positive for CML, but CML was not detected in HSA–glucose (Figure 2E). These results show that each glycation method leads to modification of HSA protein both in structural and migration profiles, but also resulting in distinct AGEs/MRPs.

### 3.3. Method of Glycation of HSA Significantly Impacts Binding to AGE Receptors

To measure the impact of structural changes caused by glycation of HSA on the binding of known AGE receptors (RAGE, Galectin-3, CD36, and SR-A1), glycated HSAs and their non-glycated HSA controls were analyzed using inhibition ELISA. Glycation of HSA significantly increased its binding to RAGE, with HSA–MGO showing the highest binding (20–40%) in a dose-response manner, followed by glucose (15–20%) and CML (15–20%) (Figure 3A).

All three methods of glycation led to a significant increase in the binding of HSA to the Galectin-3 receptor, with HSA glycated with glucose (20–30%) and MGO (40–80%) showing significant binding, although only HSA–MGO exhibited a clear dose-response effect, suggesting that the other two glycations may be less relevant (Figure 3B). Additionally, we also looked at the binding of glycated HSA to CD36 and SR-A1 receptors. Both HSA–glucose and HSA–MGO modifications showed increased binding to CD36 and SR-A1 compared to their controls. However, the percentage of inhibition did not exceed 20% indicating a lower binding affinity than for the RAGE and Gal-3 receptor (Appendix A). To identify specific fractions of glycated HSA that bind to the RAGE, a Western blot with the RAGE receptor was conducted (Figure 3C). RAGE bound to the entire HSA–glucose, including the smear bands and main HSA band. For HSA–MGO, RAGE appeared to bind specifically to the higher molecular weight proteins, presumably against aggregates. For HSA–CML, RAGE appeared to bind specifically to the main glycated albumin protein (Figure 3C). In conclusion, these data show that glycation of HSA increases the binding affinity of HSA to AGE receptors, especially RAGE and Galectin-3, with HSA–MGO showing a clear dose-dependent response.

### 3.4. Glycated HSA Leads to an Inflammatory Response in Macrophages

To correlate the receptor binding profiles with their biological activity, the impact of glycated HSA on inducing an inflammatory response in the widely accepted THP-1 macrophage cell line was studied. Incubation of THP-1 macrophages with HSA–MGO led to a 5-fold increase in NF-κB phosphorylation, while HSA–CML led to an increased trend (Figure 4). To study the effect of glycated HSA on macrophages on the gene expression level, after prior optimization (Appendix A), THP-1 macrophages were stimulated with glycated HSA for either 8 h (HSA–MGO) or 24 h (HSA–glucose and HSA–CML). Prior to running the experiment, we confirmed the *RAGE* and *Galectin-3* gene expression on differentiated THP-1 cells (Appendix A). HSA–MGO showed the highest immunogenicity, followed by HSA–glucose, and lastly HSA–CML. Stimulation of THP-1 macrophages with HSA–MGO led to a significant increase of *IL-1β*, *IL-8*, *TNFα*, and *CD86* gene expression compared to the non-treated control (Figure 5). Stimulation of THP-1 macrophages with HSA–glucose led to a significant increase in *IL-1β*, *IL-8*, and *CD86* gene expression compared to its control (Figure 5). Treatment with HSA modified with CML led to a significant increase in *IL-1β* and *RAGE* gene expression compared to the non-treated control (Figure 5). We, therefore, conclude that glycation, using the methods described in this study, increases the immunogenicity of the HSA protein, with HSA–MGO leading to the most potent effect, followed by HSA–glucose and, lastly, HSA–CML.

### 3.5. Stimulation of Primary Adherent Monocytes with Glycated HSA Leads to Inflammatory Response

To compare the results of stimulation of THP-1 macrophages with primary cells, we conducted a pilot experiment where PBMC-derived adherent monocytes isolated from 4 healthy donors were individually stimulated with glycated HSA for 3 h after initial optimization steps (Appendix A). Only HSA–CML led to a significant response with a significant upregulation of *IL-1β* (7-fold) and *IL-8* (5-fold), and an increased trend for *TNFα* was measured (Figure 6). However, a large individual donor variation is observed (Figure 6). Some donors showed high response to HSA–glucose and HSA–MGO, although when the data were averaged, no significant increases could be observed for all 3 cytokines tested. Additionally, heated albumin control also led to a significant increase in *TNFα* (Figure 6). The individual responses can be seen in Appendix A. Thus, these data show that glycated HSA, as described in this study, can lead to altered or increased immunogenicity; however, a large individual variation is observed.

To determine whether differences between THP-1 macrophages and primary adherent monocytes could be explained due to differences in receptor expression, we conducted a simple analysis of RAGE and Galectin-3 receptor expression in these cells (Figure 7). THP-1 macrophages had a significantly higher expression of RAGE compared to the primary monocytes. Additionally, an increased trend in Galectin-3 expression is observed in THP-1 macrophages (Figure 7). For the MFI graphs from Flow Jo software; see Appendix A. Thus, receptor expression differences, at least on the level of RAGE and Galectin-3, cannot explain differences between THP-1 and primary monocyte inflammatory responses. However, these data are only based on one donor and should, therefore, be interpreted with caution.

## 4. Discussion

Glycated human serum albumin (HSA) is commonly increased in diabetic conditions [5,53,54]. While it is generally thought that the increase of blood glucose leads to this glycation, studies have shown that other forms of glycation, such as that with the metabolite methylglyoxal (MGO), are also relevant in diabetic HSA glycation and pathology [3,54,55]. In the present study, we demonstrate that three different glycating metabolites, glucose, MGO, and glyoxylic acid (CML), lead to the generation of unique AGE profiles. Each profile has a distinct impact on HSA protein structure, differences in binding affinity to relevant AGE receptors, and a varying impact on the response of macrophages. Specifically, we show that HSA–MGO has the highest binding affinity to RAGE and Galectin-3, also inducing the highest inflammatory response in THP-1 macrophages.

To investigate how the different glycations impacted HSA protein, we performed a series of characterization analyses. HSA, rich in lysine and arginine residues, is highly susceptible to glycation [53,56]. Our SDS-PAGE analysis reveals that glycation increases HSA’s molecular weight and causes aggregate formation, indicated by slower migrating bands and smears in the gel. These changes confirm structural modifications, with MGO having the most significant impact, followed by glucose and then CML. Each glycation method uniquely alters the protein structure, as demonstrated by distinct gel migration profiles and LC-MS results, corroborating earlier research [3,57,58]. This protein modification is supported by other studies showing that glycation decreases alpha helices and increases beta sheets in proteins [3,54]. Notably, studies have shown that aggregates containing beta-sheet structures showed higher immunogenicity [59]. It was suggested that such structural changes can potentially affect HSA’s biological functions, such as the transport of endogenous compounds [53,56,60]. MGO levels are elevated in diseases like diabetes and age-related conditions [11]. While our in vitro samples showed a more exaggerated impact regarding aggregation, especially with MGO modification, interestingly, HSA isolated from diabetic patients was shown to aggregate similarly, likely due to MGO and glucose glycation present in vivo [61].

The most significant structural changes observed in HSA–MGO could be explained by the formation of crosslinking AGEs as a result of MGO glycation, such as methylglyoxal-lysine dimer (MOLD) [62]. MGO glycation occurs primarily on arginine residues, producing MG-H1, and secondarily on lysine residues, producing CEL (a homolog of CML) and MOLD [63]. HSA glycated with glucose also showed some increase in aggregation aligning with previous studies [60], albeit less than HSA–MGO, and may be a result of the formation of crosslinking AGEs such as pentosidine [62]. Additionally, our data showed that glycation of HSA with glucose leads to the formation of high levels of furosine, an early glycation reaction product. Regarding HSA–CML, our results showed minimal structural changes except for a slight molecular weight increase and decreased positive charge. Unlike the other glycating agents, CML alone does not cause crosslinking [3]. Thus, in summary, our results indicate that each glycation method had a unique impact on the HSA protein and generated unique Maillard Reaction Products (MRPs) and AGE profiles. HSA–glucose contained early MRPs, AGEs, and aggregates; HSA–MGO had the highest level of MG-H1, CEL, and aggregation; and HSA–CML contained no aggregates but a decreased positive charge due to the high number of CML modifications. In vivo, various glycation sites with different levels of glycation have been reported in people with diabetes [4,55,64,65,66]. Likely we do not faithfully mimic this in our in vitro study, which is a limitation. Despite this variability, there are strong similarities between the results of structural studies that have been obtained for in vivo and in vitro glycated HSA [61]. Thus, our data provides insight into the potential differential impact of the glycation with glucose or MGO.

To investigate how different glycations impact the binding of HSA to the RAGE receptor, we conducted inhibition ELISA and Western blot analysis. We demonstrated that HSA–MGO has the highest binding affinity for RAGE, consistent with previous studies [67]. RAGE, a multiligand receptor, binds AGEs and other ligands, including aggregated proteins [27,29,30,31,68]. AGEs are shown to bind to the V domain of RAGE, and it has also been shown that CEL moiety binds the positively charged cavity of the V domain [29]. Additionally, it was shown that MG-H1 peptides can bind with the V domain of RAGE via a mechanism involving MG-Hs moiety, which is different from the ε-amino group-modified lysines like CML and CEL [67]. Additionally, recent studies also show the interaction between MOLD and RAGE [63]. Together, this may explain the higher binding of HSA–MGO to RAGE as HSA–MGO. Additionally, MG-H1 was shown to be associated with deleterious effects in diabetes [11]. Indeed, RAGE Western blot results showed the binding of RAGE to higher molecular weight bands seen in HSA–MGO. In MGO glycation, serum albumin’s glycated amino acids bind directly to RAGE, primarily through dipole-dipole and charge-dipole interactions facilitated by MGO’s imidazoline ring [67]. Of note, our Western blot results showed the strongest binding of RAGE to HSA–glucose bands, which may indicate that HSA–glucose—RAGE interaction is more stable in the Western blot setup compared to our inhibition ELISA setup where the tested ligand competes with a previously established strong ligand (highly glycated soy proteins). Conversely, while HSA–MGO showed the strongest binding to RAGE in our ELISA, this binding was not as strong in the Western blot, which may be explained by receptor binding site modification by SDS. While CML is known for inducing inflammatory effects [10,69], limited research has focused on the cytotoxicity of furosine, an early MRP present in glucose glycation. Recent studies indicate the relevance of early glycated albumin in diabetes pathology, showing its specific binding to the RAGE ectodomain [60,70]. HSA glycated with glucose is strongly linked to diabetic complications [5,53], with Lys525 being the most commonly glycated residue on HSA in diabetic patients [70]. Thus, glucose glycations contain both CML and crosslinked protein, both reported to bind to RAGE, as well as a collection of other AGEs, including early and intermediate products, which are not extensively studied in relation to RAGE binding. Even though HSA–CML contained roughly 10× more CML than HSA–glucose, our ELISA results reveal similar RAGE binding intensities for both HSA–CML and HSA–glucose, which may suggest CML alone to be a weaker ligand for RAGE. However, clear binding of RAGE to HSA–CML according to our Western blot results may also suggest that it is less stable in our inhibition ELISA setup. While CML is extensively studied and associated with inflammation upon RAGE binding [10,69], some studies suggest it may not exclusively bind to RAGE [71,72] due to being very homogenous, unlike glucose and glycolaldehyde-modified proteins, as suggested by Buetler et al. [73]. Conversely, other studies show that CML can bind to RAGE via interaction involving the increased negative charge of CML-protein and positively charged cavity within the V domain of sRAGE [31], and it was shown that polyvalent engagement of CML to RAGE oligomers leads to strong binding [29]. CML-glycated serum albumin was shown to require an interaction with both CML-modified residues and the peptide backbone to bind RAGE [29]. Unlike glucose and MGO modification, CML modification does not lead to protein crosslinking and aggregation [74]. Thus, our results suggest that HSA–glucose has many potential players leading to receptor binding, including CML and CEL, whereas for HSA–MGO, the MG-H1 moiety, CEL, and aggregates are the most important players. Lastly, HSA–CML binds RAGE; however, this binding may be less stable in our inhibition ELISA setup. Additionally, when looking at other receptors, we found a dose-response increase of HSA–MGO affinity to Galectin-3 receptor. Galectin-3 was also shown to bind to aggregated proteins [25], which corresponds with our results showing HSA–MGO leading to highest aggregate formation.

Activation of AGE receptors may trigger pro-inflammatory signaling pathways [75,76,77,78,79], leading to the production of inflammatory cytokines and consequently contributing to chronic low-grade inflammation. To study how different glycations impact the immunogenicity of the HSA protein, we stimulated THP-1 macrophages with glycated vs. non-glycated HSA. HSA–MGO increased phospho-NFκB P65 activity and transcription of pro-inflammatory genes significantly, compared to other glycation methods, suggesting enhanced immunogenicity of MGO-derived AGEs. These results are in line with receptor binding profiles obtained from different glycation of HSAs. The higher affinity of HSA–MGO for the RAGE receptor correlates with elevated pro-inflammatory cytokine gene expression, indicating that higher CEL/aggregates/MG-H1 in HSA–MGO leads to higher intracellular response compared to HSA–glucose. HSA–glucose-stimulated THP-1 macrophages exhibited increased pro-inflammatory cytokine gene expression but to a lesser extent than with HSA–MGO. Previous studies also noted the upregulation of pro-inflammatory genes with HSA glycated by glucose [47]. Noteworthy, HSA–CML led only to an increase in *IL-1β* expression from all the cytokines we tested. Previous studies found a significantly higher cytokine response from macrophages stimulated with glycated albumin [34,35,36], potentially due to variations in different cell types, HSA isolation methods, use of BSA (instead of HSA), endotoxin control, or glycation techniques. Nevertheless, our findings align with this trend, indicating increased inflammatory responses post-glycation. Moreover, our results suggest a closer resemblance to the chronic inflammatory state seen in metabolic disorders like diabetes [80,81]. Extremely elevated cytokine responses, akin to those in sepsis, are less probable in conditions associated with AGEs like diabetes [2]. Furthermore, our study reveals that these types of modifications can induce a unique binding profile of modified HSA to receptors expressed on immune cells, which can have important implications for health and diabetic complications. It is important for future studies to investigate in human trials whether this can be an attributing factor to the low-grade inflammation that is reported in T2DM [80]. Thus, our data suggest that glycation can lead to inflammatory responses in patients with T2DM via mechanisms involving (early) MRPs, AGEs, and aggregates. How MGO and glucose-derived protein modification develop over the course of diabetes is hardly studied, and more attention should be given to investigating and understanding the potential contribution of these glycations to low-grade inflammation.

Lastly, to confirm if THP-1 results also translate to primary cells, we conducted a pilot experiment with primary adherent monocytes from four healthy donors. The results showed that individuals may be more reactive to glycated proteins than others, and this hypothesis requires a larger number of individuals to be tested. The large interindividual variation is in line with other metabolic studies [82,83]. This difference could be due to differences in the expression of AGE receptors. Also, the disease status can modulate RAGE and galectin-3 levels [84,85]. Accordingly, an increase in galectin-3 expression may cause less interaction between AGEs and RAGE due to competition [85], thus leading to an altered inflammatory response. Such differences can explain the individual variance in responses of allegedly healthy donors due to unknown underlying conditions such as prediabetes. Such conditions may expose monocytes to chronic low-grade inflammation, which has been shown to affect innate immunity [80,86]. CML is the most studied AGE regarding negative health impacts [10,87] and was used in this study as a reference for glucose glycation. Intriguingly, here HSA–CML appears to be the most immunogenic protein. This may be due to differences between cell lines used and the presence of receptors highly responsive to CML in primary cells not investigated in this study. Additionally, differences in RAGE oligomerization between cell lines may also explain differing outcomes [29]. CML, which is naturally a part of HSA–glucose glycation, therefore highlights the importance of glucose management to prevent CML-mediated effects. Our data show key structural and functional changes caused by the in vitro glycation of HSA, highlighting the need for future studies to explore how these changes are affected by glycation in vivo, such as in T2DM patients.

## 5. Conclusions

In conclusion, we showed that each HSA modifications lead to unique structural changes. Nevertheless, each of the modified HSA induced low inflammatory responses in THP-1-derived macrophages. HSA–MGO was the most potent modification in terms of receptor binding and induction of inflammation, which, based on biochemical analyses, might be related to the process of aggregation induced by this modification. A second potent modification, HSA–glucose, likely acts via a mechanism of action involving early MRPs or AGEs, including CML. Finally, in primary adherent monocytes, a high inflammatory response was observed after stimulation with HSA–CML. Our study aids in understanding the role of different types of diabetic glycation of HSA in the pathology of diabetes. This fundamental knowledge might help in designing HSA-focused strategies for the prevention and treatment of diabetic complications.

## Figures and Tables

**Figure 1 biomolecules-14-01492-f001:**
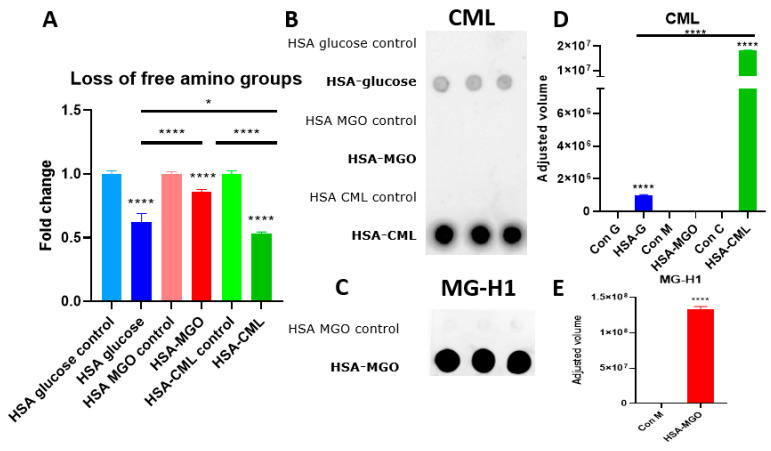
Confirmation of glycation of human serum albumin. (**A**) Loss of amino groups measured with OPA assay. (**B**,**C**) The presence of CML or MG-H1 was measured with dot blot and (**D**,**E**) quantified data of dot blot. Data presented as triplicate, 30 μg per dot. Data shown as mean ± SD of triplicate wells. Significant differences were analyzed with the Student’s *t*-test (GraphPad Prism): * *p* < 0.05, **** *p* < 0.0001.

**Figure 2 biomolecules-14-01492-f002:**
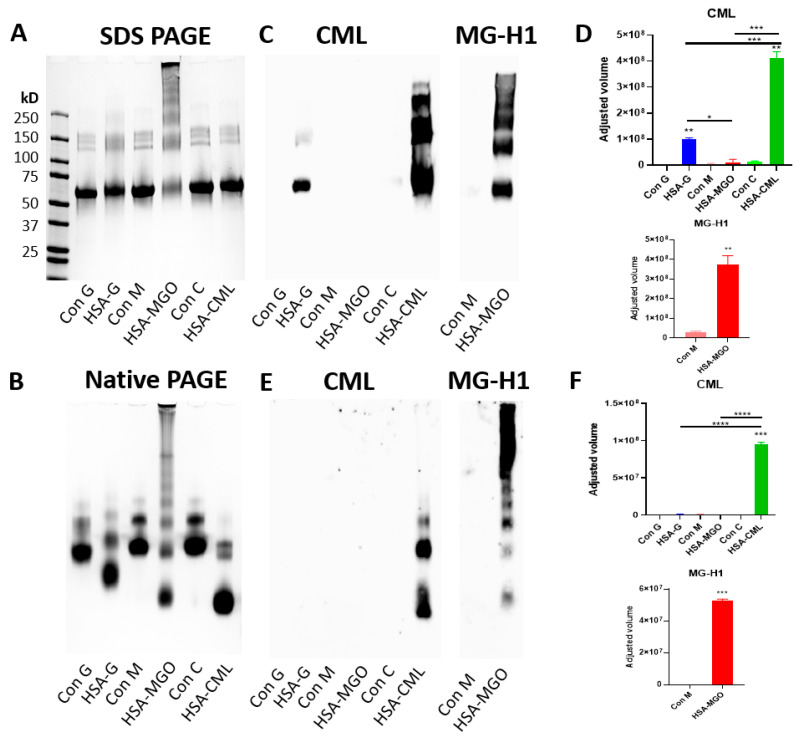
Effect of glycation on the HSA protein. A total of 8 μg of glycated HSA and non-glycated HSA controls were analyzed via SDS-PAGE (**A**) and NATIVE-PAGE (**B**). Gels were visualized with Coomassie brilliant blue staining. Additionally, a Western blot was performed after denaturing, and Native Gel electrophoresis and tested with anti-CML and anti-MG-H1 antibodies, and data were quantified for both SDS (**C**,**D**) and NATIVE (**E**,**F**) Gels. Data were analyzed using the ImageLab software by Biorad. Significant differences analyzed with the Student’s *t*-test (GraphPad Prism); * *p* < 0.05, ** *p* < 0.01, *** *p* < 0.001, **** *p* < 0.0001.

**Figure 3 biomolecules-14-01492-f003:**
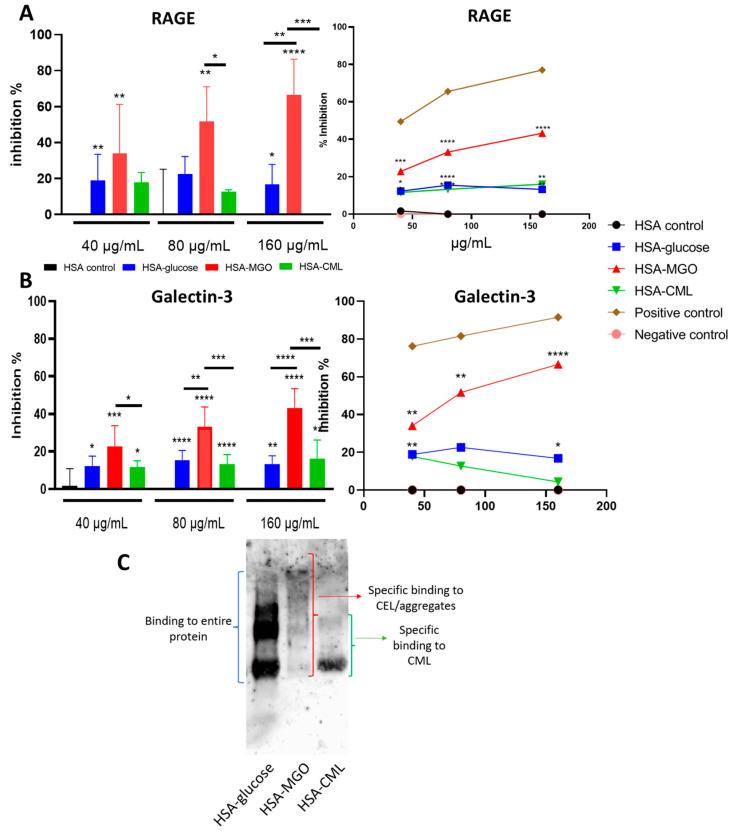
Binding of glycated HSA to AGE receptors. Glycated HSA and non-glycated HSA analyzed via inhibition ELISA for binding efficiency against (**A**) RAGE and (**B**) Galectin-3. Glycated soy and ovalbumin were used as positive and negative controls. Western blot against RAGE receptor showing to which glycated HSA RAGE binds (**C**). Data shown as mean ± SD of duplicate wells and are representative of three individual experiments. Significant differences analyzed with the Student’s *t*-test (GraphPad Prism); * *p* < 0.05, ** *p* < 0.01, *** *p* < 0.001, **** *p* < 0.0001.

**Figure 4 biomolecules-14-01492-f004:**
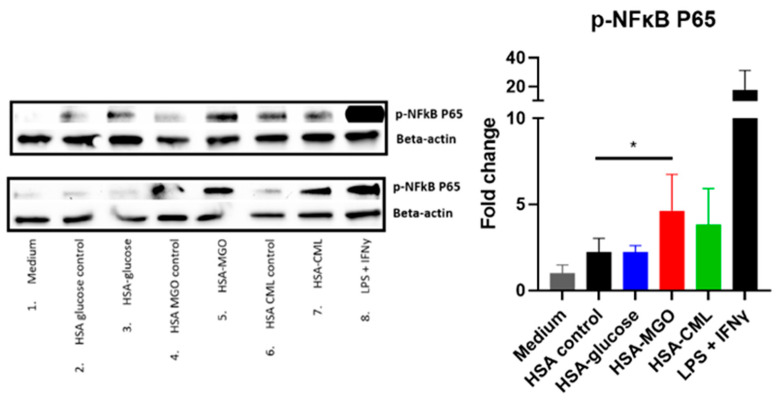
Phosphorylated NF-κB P65 protein expression in response to treatment with glycated HSA in THP-1 macrophages. THP-1-derived macrophages were stimulated with either 100 μg/mL of glycated HSA or non-glycated HSA for 10 min, followed by Western blot analyses for phosphorylated NF-κB P65 protein expression. Stimulation with LPS and IFNy was used as positive control. Data shown as mean ± SD of duplicate experiments. Western blot data quantified and analyzed using ImageLab software from Biorad. HSA controls for all 3 glycations were pooled together for quantification. Significant differences analyzed with the Student’s *t*-test (GraphPad Prism); * *p* < 0.05.

**Figure 5 biomolecules-14-01492-f005:**
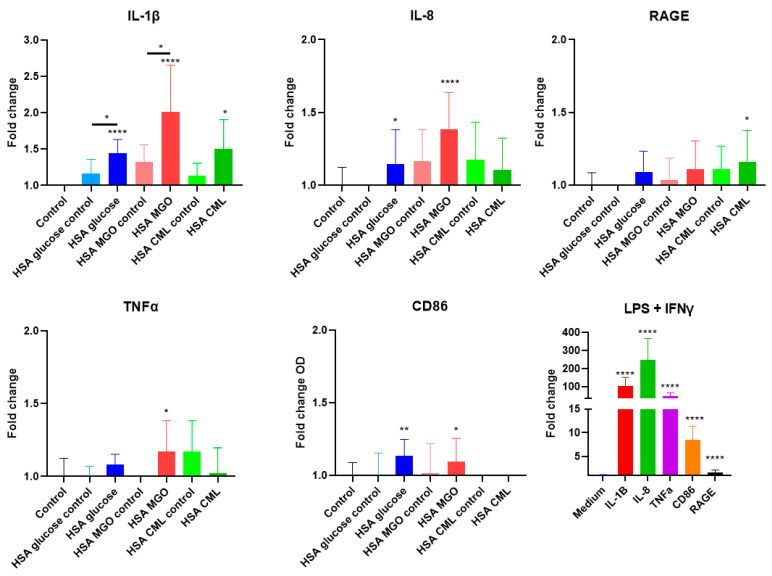
The effect of stimulation with glycated HSA on pro-inflammatory markers in THP-1 macrophages. THP-1 macrophages were stimulated with 100 μg/mL of either glycated HSA or non-glycated HSA for either 8 h (HSA–MGO) or for 24 h (HSA–glucose and HSA–CML), followed by qPCR analyses. Data shown as mean ± SD of triplicate wells and are representative of at least two individual experiments. LPS and IFNy were used as a positive control, as shown in a separate graph. Buffer controls are shown in Appendix A. Significant differences analyzed with the Student’s *t*-test (GraphPad Prism); * *p* < 0.05, ** *p* < 0.01, **** *p* < 0.0001.

**Figure 6 biomolecules-14-01492-f006:**
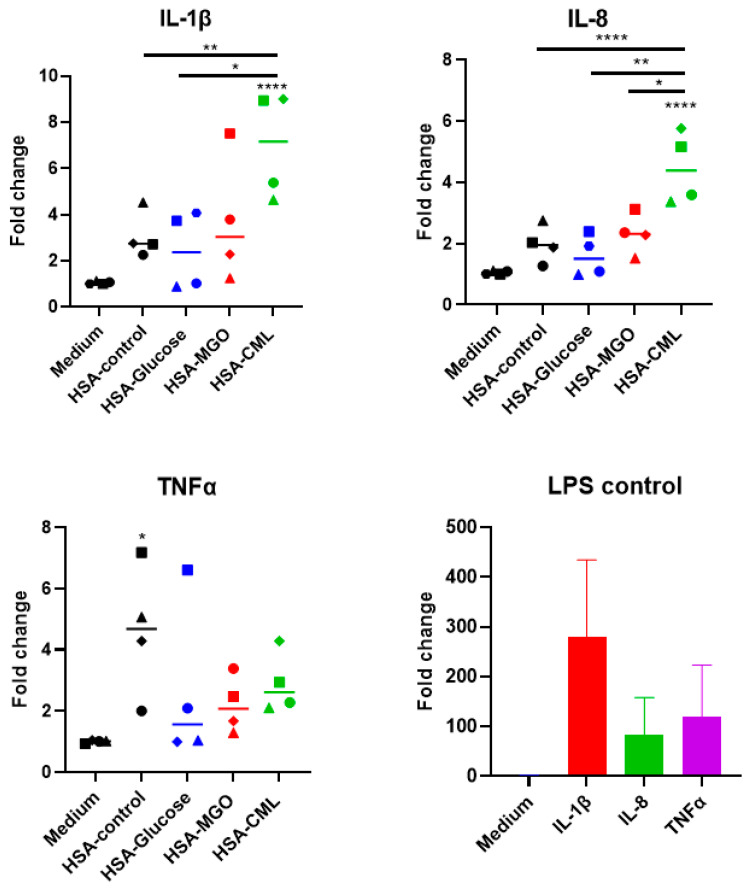
Effect of stimulation with glycated HSA on primary monocytes. PBMC-derived monocytes from 4 donors were stimulated with 100 μg/mL of either glycated HSA or non-glycated HSA for 3 h, followed by qPCR analysis of *IL-1β*, *IL-8*, and *TNFα*, and positive control LPS + IFNy showing that donors monocyte respond well to inflammatory stimuli. Data shown as mean ± SD of duplicate wells. Combined data of all donors, Significant differences analyzed with One-way ANOVA (GraphPad Prism); * *p* < 0.05, ** *p* < 0.01, **** *p* < 0.0001.

**Figure 7 biomolecules-14-01492-f007:**
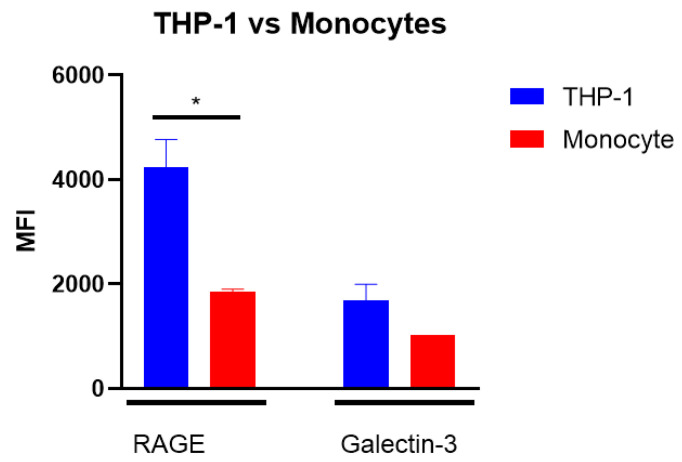
Expression of receptors on THP-1 macrophages and primary adherent monocytes. THP-1 macrophages and PBMC-derived adherent monocytes were analyzed via a Flow cytometer. Significant differences analyzed with the Student’s *t*-test (GraphPad Prism); * *p* < 0.05.

**Table 1 biomolecules-14-01492-t001:** MRPs/AGEs profile measured by LC-MS, data shown in mg/g protein.

Sample	Furosine	CML	CEL
HSA–glucose control	0.04	0.13	0.01
HSA–glucose	12.56	7.13	0.37
HSA–MGO control	0.38	0.08	0.01
HSA–MGO	0.35	0.25	3.73
HSA–CML control	0.28	0.20	0.10
HSA–CML	0.17	87.87	0.00

## Data Availability

The authors state that the data underlying the findings of this study are included in the article and its Appendix A. Raw data supporting the study’s conclusions can be obtained from the corresponding author upon reasonable request.

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
