# Peer review of "Diabetic Glycation of Human Serum Albumin Affects Its Immunogenicity"

_biomolecules, 2024, doi:10.3390/biom14121492_

Round 1
Reviewer 1 Report
Comments and Suggestions for Authors
In this manuscript, the authors show the differences between preparations of human serum albumin glycated in three different ways. The effects of these three glycation methods on albumin structure/modifications/cross-linking were demonstrated by NATIVE and SDS-PAGE, Western blotting, LC-ESI-MS/MS, through binding to receptors for advanced glycation end-products. Furthermore, inflammatory responses of macrophages and monocytes isolated from four different healthy individuals were also studied. The study is very thorough, and methodologically impeccable. The results are very significant for the understanding of the mechanisms involved in Diabetes type II. I strongly suggest publishing of this paper after the following minor corrections are amended:
Line 170 - the abbreviation PMA should be explained
Line 382 - a sentence remained from the original template
Figure 3 should be improved regarding the consistency in formatting (font size) and resolution (the figure is very blurry).
Figure 4 - it should be highlighted what was used as a HSA control on the graph - average of all of the HSA controls (glucose, MGO, CML), or something else?
Comments on the Quality of English LanguageLine 48 - there should be "and" after "healthy individuals"
Author Response
Comment 1: Line 170 - the abbreviation PMA should be explained
Response 1: This abbreviation has been modified to be explained.
Comment 2: Line 382 - a sentence remained from the original template
Response 2: This has been fixed in the manuscript.
Comment 3: Figure 3 should be improved regarding the consistency in formatting (font size) and resolution (the figure is very blurry).
Response 3: All figures will be provided separately in higher quality.
Comment 4: Figure 4 – it should be highlighted what was used as a HSA control on the graph – average of all of the HSA controls (glucose, MGO, CML), or something else?
Response 4: The caption now correctly states what the HSA-control samples represent. Indeed it is the pooled average of all the HSA-controls.
Comment 5: Line 48 – there should be “and” after “healthy individuals”
Response 5: This modification has been made in the revised manuscript.
Reviewer 2 Report
Comments and Suggestions for Authors
Manuscript submitted by Croes et. al "Diabetic glycation of human serum albumin affects its immunogenicity" in the biomolecules (biomolecules-3265008). Revision is required before the publication of this manuscript after including these comments. 1. Author should provide high quality image. 2. Diabetes Mellitus should be “diabetes mellitus” 3. Carefully check the abbreviations in the manuscript to ensure their consistency. 4. Chemicals name should be standard format e.g NaBH3CN. 5. Author can take help from this article in the introduction section ‘Int J Biol Macromol. 2021, 193, 948-955’. 6. Conclusion section is not properly correlated with current finding and how this study beneficial for future studies.
Comments on the Quality of English LanguageNo
Author Response
Comment 1: Author should provide high quality image.
Response 1: All figures will be provided separately in high quality.
Comment 2: Diabetes Mellitus should be “diabetes mellitus”
Response 2: Thank you, this has been modified in the revised manuscript.
Comment 3: Carefully check the abbreviations in the manuscript to ensure their consistency.
Response 3: The manuscript was checked and minor revisions were made throughout the manuscript to ensure consistency in abbreviation.
Comment 4: Chemicals name should be standard format e.g NaBH3CN.
Response 4: Modification has been made in the manuscript, also for water, hydrochloride and acetonitrile.
Comment 5: Author can take help from this article in the introduction section ‘Int J Biol Macromol. 2021, 193, 948-955’
Response 5: This paper is now cited at line 63 of the revised manuscript to explain basic functions of HSA that may be affected by glycation. Thank you for suggesting this paper, as it also made us realise that the basic function of HSA was lacking in the original introduction.
Comment 6: Conclusion section is not properly correlated with current finding and how this study beneficial for future studies.
Response 6: The conclusion section was revised to correlate better with our results and to explain more clearly how our findings may aid future studies.
Reviewer 3 Report
Comments and Suggestions for Authors
The manuscript entitled “Diabetic glycation of human serum albumin affects its immunogenicity” addresses a very important question regarding hyperglycaemia related toxicity. Protein glycation is one of the main axes of glucose toxicity in diabetes, and while it has deserved high attention more than a decade ago, the interest of the biomedical community has fainted in recent years. Particularly when addressing the relevance and nuances of different AGEs or considering the importance of structural aspects of the toxicity mechanism. Therefore, I believe the manuscript addresses a significant problem and overlall presents a sound experimental design to address the immunogenic impact of some of the most relevant AGE structures occurring in HSA.
I do however have a major question regarding the preparation of glycated HSA samples which are used in subsequent experiments, which I believe that needs to be addressed, in order to allow the authors to obtain physiologically meaningful results.
Protein modification was obtained under very drastic conditions, high concentrations of glycating reagents and long incubation time. The authors cite that under a diabetic condition up to 50% of HSA may be glycated, but this usually corresponds to the site occupancy at a single lysine residue (K545), while for their samples an average of 40% of all 59 lysines/per HSA molecule are modified by glucose. Similar conclusions can be drawn for the other modifications. This elevated level of modification is likely to have a very extensive impact on the overall protein structure and charge balance, which is not likely to occur in vivo to the same extent. It is well reported that the half life of HSA is diminished in diabetes as a consequence of glycation, and therefore, the accumulation of so many non-enzymatic PTMs is not likely to occur. Furthermore, results in figure 2 clearly show that modification with glyoxal and glyoxylic acid induced extensive protein aggregation, which is not easily observed in the blood plasma of diabetes patients. Cross-links can be detected, but they occur at trace levels.
This extensive aggregation, change in overall charge and likely disorder of tertiary structure, will profoundly impact the interaction of modified HSA with RAGE and scavenger receptors, playing a major role in the outcome of cell culture experiments.
My recommendation is for the authors to provide a major revision of the manuscript, where they can demonstrate that this conclusions hold for modified HSA obtained under conditions which are able to reproduce glycated-HSA structures which are more to occur in vivo.
Author Response
Comment 1: My recommendation is for the authors to provide a major revision of the manuscript, where they can demonstrate that this conclusions hold for modified HSA obtained under conditions which are able to reproduce glycated-HSA structures which are more to occur in vivo.
Response 1: This is a very good point raised by the reviewer. Each point is addressed one by one below and the discussion has been revised in lines 536-539, 555-560 and conclusion part.
Comment 2: “Protein modification was obtained under very drastic conditions, high concentrations of glycating reagents and long incubation time. The authors cite that under a diabetic condition up to 50% of HSA may be glycated, but this usually corresponds to the site occupancy at a single lysine residue (K545), while for their samples an average of . Similar conclusions can be drawn for the other modifications. This elevated level of modification is likely to have a very extensive impact on the overall protein structure and charge balance, which is not likely to occur in vivo to the same extent.”
Response 2: Regarding the high concentrations and long incubation times, it is true that a higher concentration was used for both MGO and glucose to ensure high glycation levels which gave higher responses in our mechanistic studies. However only modification with glucose under went a long incubation time while MGO glycation occurred overnight. Both samples were then filtered to remove residual glycating agents.
With the OPA assay used in this manuscript, we show the percentage of blocked or modified amino groups in general not restricted to lysine. Therefore, it is difficult to correlate our data with percentage of glycation of the lysine residue specifically. For this a much more advanced proteomic assay should be used which was out of the scope of this study but it is included in the plan of a follow up project. Those techniques include MS/MS and bottom up proteomics.
When looking into the literature, lysine residue 525 is the predominant residue that is more commonly associated with in vivo glucose glycation in diabetics. Regarding the percentage however, what is reported is mixed. Indeed some studies reported up to 90% of the glycation occurs on this site in severe diabetics(Kisugi et al., 2007), and that glycation at this site alone also increased binding to RAGE receptor(Tramarin et al., 2020). While other reports still name lysine 525 as the predominant in vivo glycation site, they also report the glycation percentage at this site to be closer to 30%(Iberg & Flückiger, 1986; Kisugi et al., 2007). Interestingly, a study by Kisugi et al showed that HSA from a severe diabetic patient showed 26.4% of K525 glycated, and for normal diabetic 38.9% of K525 glycated, in addition to other sites(Kisugi et al., 2007). While it is true that in vitro glycation allows for glycation at more sites(Kumari et al., 2021), other studies suggest that in vivo glycation also encompasses other lysine groups(Anguizola et al., 2013). Table 1 in the review by Anguizola et al shows a nice overview of other lysine residues also modified in vivo(Anguizola et al., 2013). Regarding in vitro glycation, some studies found overlapping characteristics such as lysine residue 525 being primary glycating site(Barnaby et al., 2011). Another study by Duerin-Dubourg et al also showed comparable MRPs in diabetic HSA vs in vitro glycated HSA, but noted that diabetic glycated HSA actually had more drastic functional outcomes than in vitro glycation, and in that sense in vivo glycated diabetic HSA remains the most relevant model to study(Guerin-Dubourg et al., 2012).
Thus, when taking all of this information together, in vitro glycation differs from in vivo glycation, however since in vivo glycation also encompasses multiple and similar glycation sites, our in vitro models can still deliver meaningful information and aid in the understanding the mechanisms leading to negative health outcome, especially by separating glucose and MGO glycation leading to more mechanistic analysis.
The limitation that in vivo responses may differ from the in vitro responses here has now been added to the discussion in the revised manuscript. Thus a summary of this section has been added to lines 555-560 with the following “In vivo various glycation sites with different levels of glycation have been reported in diabetics[4, 56, 65–67]. Likely we do not faithfully mimic this in our in vitro study, which is a limitation. Despite this variability, there are strong similarities between the results of structural studies that have been obtained for in vivo and in vitro glycated HSA[62]. Thus our data provides insight in the potential differential impact by the glycation with glucose or MGO”. Additionally, a sentence has been added to the material and methods to address that higher concentration of glycation agents were used to assure maximum glycation at line 118-119.
Comment 3: “It is well reported that the half life of HSA is diminished in diabetes as a consequence of glycation, and therefore, the accumulation of so many non-enzymatic PTMs is not likely to occur.”
Response 3: The major goal of this study was to investigate the effect of HSA modification on receptor binding and activation of pro-inflammatory responses. This is a pure consequence of AGEs, which are constant in people with diabetes, as well as aggregation of glycated HSA, as a direct consequence of glycation(Guerin-Dubourg et al., 2012), rather than a consequence of accumulation of MRPs. Accumulation of AGEs will play a role but rather in local tissue based response. Therefore, the half-life of HSA according to our knowledge in relation to our research questions will not be a major factor influencing our outcomes. To avoid confusion, the word “accumulation” has been removed from the manuscript, as this was only mentioned twice. Finally it is not clear that glycation shortens half-life of HSA. While a reduction has been reported(Nakajou et al., 2003), a drastic diminished half-life is not clearly stated in the literature. In fact, some studies still report similar half-life of glycated HSA as non-glycated HSA, namely 3 weeks(Baynes et al., 1984). Additionally, particularly due to this half-life efforts have been made to use glycated HSA as a diabetic screening tool to screen 2-3 weeks of glycemic activity(Raghav & Ahmad, 2014; Zendjabil, 2020).
Comment 4: Furthermore, results in figure 2 clearly show that modification with glyoxal and glyoxylic acid induced extensive protein aggregation, which is not easily observed in the blood plasma of diabetes patients. Cross-links can be detected, but they occur at trace levels. This extensive aggregation, change in overall charge and likely disorder of tertiary structure, will profoundly impact the interaction of modified HSA with RAGE and scavenger receptors, playing a major role in the outcome of cell culture experiments.
Response 4: In figure 2, only glucose and methylglyoxal modification (HSA-glucose and HSA-MGO) led to aggregation, not glyoxylic acid modification (HSA-CML). As we have cited in the manuscript, glucose and methylglyoxal modification leads to AGEs that are known to crosslink, whereas CML does not crosslink. A study by Guerin-Dubourg nicely showed an SDS page with HSA from diabetic patient showing clear signs of protein aggregation, similar to what we show in Figure 2. It is known that HSA glycation in diabetics can occur via mechanisms involving both glucose and methylglyoxal, so the data from Guerin-Dubourg et al reflects both of these glycations occurring within the diabetic HSA sample(Guerin-Dubourg et al., 2012). Indeed, our samples shows more exaggerated variants of glucose and methylglyoxal glycation, however by separating these methods and studying them as we did in our manuscript we further increase our understanding of which modification is more detrimental to health. Our data clearly reveals methylglyoxal modification having higher potential to cause aggregation compared to glucose. A summary of this section has been added to the discussion at lines 536-538, “While our in vitro samples showed a more exaggerated impact regarding aggregation, especially with MGO modification, interestingly HSA isolated from diabetic patients was shown to aggregate in a similar manner, likely due to MGO and glucose glycation present in vivo(Guerin-Dubourg et al., 2012).”
Regarding the binding to the receptors, indeed aggregation plays a part in the binding to receptors, but it is not the sole property leading to the binding and activation of these receptors. As cited in our discussion, different molecular aspects of glycation also leads to binding of these receptors, such as mechanisms involving single CEL or CML moieties, or MG-H1 peptides to the V domain of RAGE. Thus RAGE ligands have different sizes ranging all the way from aggregates to small molecules.
References
Anguizola, J., Matsuda, R., Barnaby, O. S., Hoy, K. S., Wa, C., DeBolt, E., Koke, M., & Hage, D. S. (2013). Review: Glycation of human serum albumin. Clinica Chimica Acta, 425, 64–76. https://doi.org/10.1016/j.cca.2013.07.013
Barnaby, O. S., Cerny, R. L., Clarke, W., & Hage, D. S. (2011). Comparison of modification sites formed on human serum albumin at various stages of glycation. Clinica Chimica Acta, 412(3), 277–285. https://doi.org/10.1016/j.cca.2010.10.018
Baynes, J. W., Thorpe, S. R., & Murtiashaw, M. H. (1984). [8] Nonenzymatic glucosylation of lysine residues in albumin. In Methods in Enzymology (Vol. 106, pp. 88–98). Academic Press. https://doi.org/10.1016/0076-6879(84)06010-9
Guerin-Dubourg, A., Catan, A., Bourdon, E., & Rondeau, P. (2012). Structural modifications of human albumin in diabetes. Diabetes & Metabolism, 38(2), 171–178. https://doi.org/10.1016/j.diabet.2011.11.002
Iberg, N., & Flückiger, R. (1986). Nonenzymatic glycosylation of albumin in vivo. Identification of multiple glycosylated sites. Journal of Biological Chemistry, 261(29), 13542–13545. https://doi.org/10.1016/S0021-9258(18)67052-8
Kisugi, R., Kouzuma, T., Yamamoto, T., Akizuki, S., Miyamoto, H., Someya, Y., Yokoyama, J., Abe, I., Hirai, N., & Ohnishi, A. (2007). Structural and glycation site changes of albumin in diabetic patient with very high glycated albumin. Clinica Chimica Acta, 382(1), 59–64. https://doi.org/10.1016/j.cca.2007.04.001
Kumari, N., Bandyopadhyay, D., Kumar, V., Venkatesh, D. B., Prasad, S., Prakash, S., Krishnaswamy, P. R., Balaram, P., & Bhat, N. (2021). Glycation of albumin and its implication in Diabetes: A comprehensive analysis using mass spectrometry. Clinica Chimica Acta, 520, 108–117. https://doi.org/10.1016/j.cca.2021.06.001
Nakajou, K., Watanabe, H., Kragh-Hansen, U., Maruyama, T., & Otagiri, M. (2003). The effect of glycation on the structure, function and biological fate of human serum albumin as revealed by recombinant mutants. Biochimica et Biophysica Acta (BBA) - General Subjects, 1623(2), 88–97. https://doi.org/10.1016/j.bbagen.2003.08.001
Raghav, A., & Ahmad, J. (2014). Glycated serum albumin: A potential disease marker and an intermediate index of diabetes control. Diabetes & Metabolic Syndrome: Clinical Research & Reviews, 8(4), 245–251. https://doi.org/10.1016/j.dsx.2014.09.017
Tramarin, A., Naldi, M., Degani, G., Lupu, L., Wiegand, P., Mazzolari, A., Altomare, A., Aldini, G., Popolo, L., Vistoli, G., Przybylski, M., & Bartolini, M. (2020). Unveiling the molecular mechanisms underpinning biorecognition of early-glycated human serum albumin and receptor for advanced glycation end products. Analytical and Bioanalytical Chemistry, 412(18), 4245–4259. https://doi.org/10.1007/s00216-020-02674-w
Zendjabil, M. (2020). Glycated albumin. Clinica Chimica Acta, 502, 240–244. https://doi.org/10.1016/j.cca.2019.11.007
Round 2
Reviewer 3 Report
Comments and Suggestions for Authors
The authors have chosen to not address my comments in full and to not provide experimental evidence that HSA glycation occurring at levels more akin to those found in DM patients still presents an immunogenic character. Arguments in support of their position are presented in their response to reviewers. I maintain my disagreement with some of their responses, and I would say that some aspects, in my view, are miss interpretations or overinterpretations of the similarities of what is observed in vivo and their in vitro generated glycated albumin.
As signaled by the authors “each of modified HSA induced low inflammatory responses in THP-1 derived macrophages”, even when the modified HSA was obtained “with high concentrations of glycating agents to assure maximum glycation”. The significance of these results under physiological conditions remains a question.
However, and despite the importance of the peer review process, it has been my ethical position that a paper should reflect the views of their authors and not those of the reviewers. Providing that there are not clear overclaims, signs of scientific miss conduct or erroneous experimental design. Furthermore, I must highlight that the paragraph introduced by the authors in the discussion section at least brings the main limitation of the work to the readers’ attention.
In my view, as it is, the manuscript has limited interest, although having the merit to address a very important question. I also believe that overall, this is a good experimental work and that it can be left to the informed reader to judge on the relevance of the conclusions.
Therefore, based on the overall relevance of the theme and not having any doubt on the quality of the data presented I recommend that the manuscript should be published.
Author Response
Comment from the reviewer:
The authors have chosen to not address my comments in full and to not provide experimental evidence that HSA glycation occurring at levels more akin to those found in DM patients still presents an immunogenic character. Arguments in support of their position are presented in their response to reviewers. I maintain my disagreement with some of their responses, and I would say that some aspects, in my view, are miss interpretations or overinterpretations of the similarities of what is observed in vivo and their in vitro generated glycated albumin.
As signaled by the authors “each of modified HSA induced low inflammatory responses in THP-1 derived macrophages”, even when the modified HSA was obtained “with high concentrations of glycating agents to assure maximum glycation”. The significance of these results under physiological conditions remains a question.
However, and despite the importance of the peer review process, it has been my ethical position that a paper should reflect the views of their authors and not those of the reviewers. Providing that there are not clear overclaims, signs of scientific miss conduct or erroneous experimental design. Furthermore, I must highlight that the paragraph introduced by the authors in the discussion section at least brings the main limitation of the work to the readers’ attention.
In my view, as it is, the manuscript has limited interest, although having the merit to address a very important question. I also believe that overall, this is a good experimental work and that it can be left to the informed reader to judge on the relevance of the conclusions.
Therefore, based on the overall relevance of the theme and not having any doubt on the quality of the data presented I recommend that the manuscript should be published.
Response
We would like to thank the reviewer for the constructive criticism which we appreciate. We are glad to learn that the reviewer appreciates our perspective and experimental design of this study. We admit that the results obtained in this study should be confirmed by a human study to confirm their in vivo relevance. We would like to explain that it is very difficult to quickly provide experimental evidence that HSA glycation occurring in vivo in the DM patients still presents clear immunogenic characteristics since it requires such a human study aiming at the isolation and chemical characterization of HSA isolated from different target groups. We would like to share with the reviewer that this is a topic of our follow up study which, if granted, will be performed in a near future. Therefore the current study , from our perspective, gives an important introduction to this human study by providing the complete characterization of different profiles of glycation of HSA and the role of compounds like MGO and CML in the immunogenicity of glycated HSA.
Regarding the manuscript revision, although we previously updated it to include statements acknowledging that our samples may not reflect in vivo glycation, as suggested by the reviewer, it is true that our conclusion from the THP-1 and monocyte experiments still did not clearly emphasize the need for cautious interpretation. Thus, in light of the recent comments from the reviewer, we have now included 3 statements which highlights that the reader of the article should interpret the data with caution, as our mechanistic study, while being highly relevant and informative, may not be fully in line with in vivo measurements.
First, in the results, at line 457 we now wrote “We therefore conclude that glycation, using the methods described in this study, increases the immunogenicity of the HSA protein”.
Secondly, in the results, at line 489, we now wrote “Thus, this data shows that glycated HSA as described in this study can lead to an altered or increased immunogenicity”
Finally, in line 662 in the Discussion, we added the following sentence “Our data show key structural and functional changes caused by the in vitro glycation of HSA, highlighting the need for future studies to explore how these changes are affected by glycation in vivo, such as in T2DM patients.”